

**Correspondence Authors:**
**Name:** Guofeng Zhu [a, b,] *
**Institution:**
[a] *College of Geography and Environment Science, Northwest Normal University,*
*Lanzhou 730070, Gansu, China*
[b] *Shiyang River Ecological Environment Observation Station, Northwest Normal*
*University, Lanzhou 730070, Gansu, China*
**Address:**
College of Geography and Environment Science of Northwest Normal
University, 967, East Anning Road, Lanzhou, Gansu, China 730000.
**Tel:** +86-13909310867
**Fax:** +86-09317971565
**E-mail:** zhugf@nwnu.edu.cn



# Evaporation loss estimation of the river-lake

# continuum of arid inland river: Evidence from stable

# isotopes

Guofeng Zhu[a,b,*] (zhugf@nwnu.edu.cn), Zhigang Sun[a,b] (zachsuen@163.com), Yuanxiao

Xu[a,b] (Xyxange@163.com), Yuwei Liu[a,b] (liuyuweinwnu@163.com), Zhuanxia Zhang[a,b]

(zzx_nwnu@163.com), Liyuan Sang[a,b] (nwnusly@163.com), Lei Wang[a,b]

(wlxbsd02468@163.com)

[a] *College of Geography and Environment Science, Northwest Normal University, Lanzhou 730070,*

*Gansu, China*

[b] *Shiyang River Ecological Environment Observation Station, Northwest Normal University,*

*Lanzhou 730070, Gansu, China*

*[*] Corresponding author*

**Abstract:** Stable isotopes could be used as tracers to estimate the evaporation loss of

surface water, which provides a new perspective for the research of hydrological

processes. A systematic observation station has been built in the Shiyang River Basin,

one of the most important inland river basins in Northwest China. This work

conducted systematic observations on the river water, precipitation and

hydrometeorology of the Shiyang River from 2017 to 2019. The evaporation loss of

the Shiyang River is estimated to be 1.30% in the mountainous rivers, 2.28% in the

mountainous reservoir (Xiying Reservoir), 2.87% in the oasis rivers, 7.97% in the

oasis reservoir (Hongyashan Reservoir), and 41.37% in the terminal lake (Qingtu



Lake). No matter in mountain or oasis, the evaporation loss of reservoir is much
higher than that of the river, and the evaporation loss of the terminal lake is the largest.
The evaporation loss of the river-lake continuum accounts for 14.66% of the total
water volume of rivers and lakes (reservoirs). This work enriches the study of stable
isotopes in the field of evaporation loss in the river-lake continuum, expands our
understanding of the hydrological cycle in arid regions.
**Keywords:** Stable isotopes, *Hydrocalculator*, Evaporation loss, River-lake continuum
**1. Introduction**
Rivers are an important path of the global water cycle and play an important role
in transporting materials and energy worldwide (Christophe et al., 2020). River water
is the most important freshwater resource that humans can be directly utilized, and it
plays a vital role in human life and the development of industry and agriculture
(Diamond and Jack, 2018). Especially for arid and semi-arid regions, river water
provides most of the production and domestic water for local residents. However, the
limitation of global warming on the efficient use of water resources by humans is
increasingly large. Limited water resources have gradually become obstacles to local
economic development, especially in arid and semi-arid areas. Therefore, alleviating
the restriction of limited water resources on social and economic growth has become a
hot topic.
Evaporation and transpiration play an essential role in the global hydrological
cycle (Brutsaert, 1986; Dogramaci et al., 2012), while evaporation tends to be the



largest contributor to continental water flux in arid and semi-arid areas due to the
sparse vegetation (Jasechko et al., 2013). Evaporation is the primary water losses for
surface water in arid and semi-arid areas (Dogramaci et al., 2012; Wang et al., 2016),
in which a significant fraction of lakes' storage (30%-50%) has evaporated
(Maestre-Valero et al., 2013; Majidi et al., 2015). Evaporation losses amounted to
40%-60% of the reservoir output and 61% of agricultural use in Texas (Katja et al.,
2018; Wurbs and Ayala, 2014), 20% of the country Nile share in Egypt (El-Shirbeny
and Abutaleb, 2018), and 40% of reservoir storage in Northwest Xinjiang in China
(Shi et al., 2016) and Queensland in Australia (Craig et al., 2005). Evaporation losses
will increase with the increasingly higher temperature (Maestre-Valero et al., 2013).
By 2100, the evaporation losses are estimated to increase by 1.09 to 2.74 mm per year,
thereby reducing the available surface water in the dry season by 5.5% to 10.4%
(Althoff et al., 2020; Zhao and Gao, 2019). Such evaporation loss leads to loss of
storage water without use for domestic, irrigation or agricultural purposes. Therefore,
as the main water resources in arid and semi-arid regions, it is necessary to
systematically assess the evaporation loss of river systems in the context of constant
climate change.

As two of the constituent elements of river water and an excellent natural tracer,

stable isotopes ($^{18}$O and $^{2}$H) provide a simple and reliable tool for estimating of river
water evaporation loss (Halder et al., 2015). Numerous studies have successfully
estimated the evaporation loss of the large open waters using stable isotopes (Cui et



al., 2017; Hernández-Pérez et al., 2020; Yapiyev et al., 2020), while few studies have
focused on mobile water systems, such as natural rivers (Diamond and Jack, 2018)
and artificial waterways (Chen and Tian, 2021). It may be that the flowing water
system will bring some instability factors, such as the discharge from reservoirs or
lakes (Luc and Bernhard, 2007; Aravena and Suzuki 1990), the inflow of important
tributaries (Wu et al., 2018; Simpson and Herczeg, 1991), the exchange between
groundwater and surface water (Winston and Criss, 2003), the return flow of
irrigation water (Mohammed et al., 2016), the inconsistency between sampling
schedule and flow velocity (Chen and Tian, 2021), the influence of rainfall during the
sampling period (Skrzypek et al., 2015). Although there are some uncontrollable
factors in the estimation of evaporation loss of the river system, it is necessary to
carry out relevant research because surface river system is the main contributor to the
surface evaporation (Chen and Tian, 2021).

As one of the most important inland rivers in Northwest China, the Shiyang

River plays an important role in the production and life of local residents and social
development. Previous studies on the Shiyang River Basin mainly focused on water
vapor recycling (Li et al., 2016; Zhu et al., 2019), soil water evaporation (Yong et al.,
2020), the interaction between groundwater and surface water (Ma et al., 2005), and
plant water sources (Zhang et al., 2021). Although the study has paid attention to the
evaporation of the Shiyang River Basin, it focused on the evaporation loss of stable
isotopes while not the surface water (Sun et al., 2021). The objectives of this work are



(a) to estimate the evaporation loss of surface water in the different river and lakes
sections; (b) to analyze the factors that cause differences in evapotranspiration loss
across the different river and lake sections; (c) to discuss uncertainties in estimating
evaporative losses from rivers and lakes using isotopic data. This study provides a set
of feasible observation and calculation schemes for the estimation of evaporation loss
in the basin.

## 2. Study area

As one of the most important inland rivers in Northwest China, the Shiyang

River (Fig. 1) has an indelible effect on the vigorous development of the Hexi
Corridor, China. Originating in the Qilian Mountains in the south and disappearing in
the Tengger Desert in the north, it covers a distance of 260 km (Shi et al., 2002). From
south to north, the elevation of the Shiyang river gradually decreases. According to
the altitude difference, Shiyang River Basin can be divided into two portions: the
Qilian Mountains in the upper reaches with an elevation of 2,000 to 5,000 m and an
alluvial plain in the middle and lower reaches with a peak of 1,300 to 2,000 m (Gao et
al., 2016).

Due to the unique geographical location of the Shiyang River in the transitional

zone between the eastern monsoon zone and the western arid zone in China, it is
affected by the East Asian monsoon and Westerly Winds (Chen et al., 2008).
Therefore, precipitation presents noticeable seasonal changes. Precipitation in the
basin is high in the summer when the East Asian monsoon has a greater impact.





Almost 70% of the annual rainfall is concentrated from June to September (Zhu et al.,
2019). Besides, the seasonality of precipitation in this area has a high degree of spatial
heterogeneity, generally decreasing along the Shiyang River pathway with the mean
annual precipitation ranging from 200 to 700 mm in the southern mountainous region,
from 150 to 300 mm in the middle oasis region, and less than 100 mm in the northern
desert region (Sun et al., 2021). Potential evaporation generally exceeds precipitation
in this area, and especially in the lower reaches of desert areas, which can reach 2,600
mm (Ma et al., 2012). The temperature gradually increases from mountain to desert,
with an annual average temperature of 9.20℃ in the mountainous area, 10.35℃ in
the oasis area, and 12.00 ℃ in the desert area. The relative humidity gradually
decreases from mountainous to the desert, with an annual average relative humidity of
51.50% in the mountainous area, 49.37% in the oasis area, and 40.21% in the desert
area, respectively. The temperature is higher in the summer and lowers in the winter.
However, there is no obvious seasonal variation in relative humidity, but high relative
humidity is highly correlated with the occurrence of precipitation. Agriculture and
animal husbandry in the middle and lower reaches also consume a lot of water,
making this area one of the regions in the world where water supply and demand are
highly imbalanced.



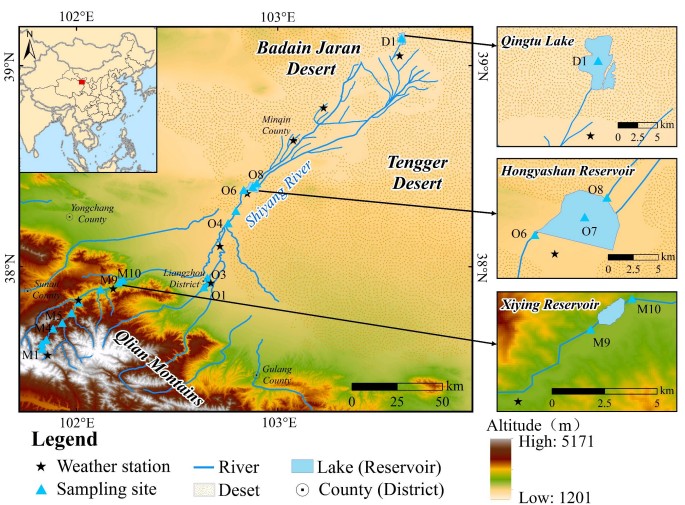


Fig. 1 Topographical overview of Shiyang River Basin and locations of the surface water
sampling sites and automatic weather stations. The digital elevation data of this map is provided
by Geospatial Data Cloud site, Computer Network Information Center, Chinese Academy of
Sciences (http://www.gscloud.cn), and used under a Creative Commons license.
## 3. Materials and methods
### 3.1 Sampling design
This study investigates the stable isotopic composition of event-based
precipitation and monthly surface water samples taken in the Shyiang River basin
from April to October in the period from 2017 to 2019. The data has been examined
previously by (Sun et al., 2021). Since the mountainous area in the upper reaches of
the Shiyang River is the main source of runoff recharge, we set up 7 surface water
observation stations in the mountainous area to more carefully detect the stable
isotopic changes of surface water in the mountainous area. 5 surface water
observation stations were also set up in natural rivers in the oasis area. In the desert

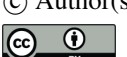



area, 1 systematic observation station, including 10 sampling points, was set up in
Qingtu Lake, the end of Shiyang River. In addition, we set up 2 reservoir observation
stations, including Xiying Reservoir located in the river exit of Qilian Mountain and
Hongyashan Reservoir located in the middle of the oasis area. The reservoir
observation system includes inlet water, reservoir water and outlet water. From April
to October during the period of 2017 to 2019, systematic sampling campaigns were
conducted once a month from upstream to downstream along the Shiyang River.
From upstream to downstream, a total of 8 precipitation observation stations
have been established, including 4 in the mountainous area, 3 in the oasis area, and 1
in the desert area. The collection of precipitation samples was based on the rainfall
event. Once the rainfall stops, the water in the rain cylinder is immediately transferred
to the standard sample bottle, labeled with time and place, and put in the refrigerator.
The meteorological data for this study were obtained from 8 automatic weather
stations which were located in the same place as precipitation observation stations.
These stations record meteorological data every 30 minutes, including temperature,
relativity humidity, dew point temperature.
**3.2 Stable isotope analysis**
Stable isotopes of oxygen ($^{18}O$ and $^{16}O$) and hydrogen ($^{2}H$ and $^{1}H$) were analyzed
in the Stable Isotope Lab of Northwest Normal University. Oxygen and hydrogen
isotope ratios were measured using the Liquid Water Isotope Analyzer (DLT-100, Los
Gatos Research, USA). Every water sample and isotope standard sample was
continuously injected six times. To avoid the memory effect of isotope analysis, the
first two injections were discarded and the average value of the last four injections
was used in the examinations (Zhu et al., 2019). The isotopic composition of oxygen
and hydrogen are reported in terms of delta ($\delta$) notation in per-mil (‰, parts per
thousand) and defined as follows:
$$\delta_{sample}(‰) = (\frac{R_{sample}}{R_{standard}} - 1) \times 1000 \qquad (1)$$
where $R_{sample}$ is the ratio of $^{18}O/^{16}O$ or $^{2}H/^{1}H$ in the samples and $R_{standard}$ is the ratio of
$^{18}O/^{16}O$ or $^{2}H/^{1}H$ in Vienna Standard Mean Ocean Water (V-SMOW). Repeated
analyses of internal standards provide an analytical precision of ±0.2‰ for oxygen
and ±0.6‰ for hydrogen, respectively (Wang et al., 2016).

**3.3 Craig-Gordon model**

Craig and Gordon (1965) developed a conceptual model that describes the stable
isotope evolution of open surface water during evaporation, which was used to solve
the evolution of stable isotopic compositions of ocean surface water during the
evaporation process. Based on the C-G model and previous researchers' verification
of this model (Gibson et al., 2002; Gibson and Edwards, 2002; Horita and
Wesolowski, 1994), Skrzypek et al. (2015) proposed the *Hydrocalculator* that allows
the estimation of evaporation losses based on the stable isotopic composition of the
input and output water of a pool and the local precipitation. It procedurals all
calculation steps, and researchers only need to input relevant parameters to get the
final calculation results, which makes the researcher's work more concise. Currently,





this software is open to all researchers on http://hydrocalculator.gskrzypek.com.
Based on the distribution of reservoirs in the Shiyang River Basin and the
sampling plan, we divide Shiyang River into seven sections (including 2 in the
mountainous areas, 2 in the oasis areas, 1 in the desert areas and 2 reservoirs), and the
evaporation loss of each section is calculated based on the *Hydrocalculator* software.
Due to the constant inflow or outflow of river water in these sections, the steady-state
model should be selected for the calculation of evaporation loss at each section.
Therefore, we should select the value of columns *EI_H* and *EI_O* in the *output* file as
the estimation results of the evaporation loss in these sections.
According to the runoff of each section, which is obtained in the Water
Resources Utilization Center of Shiyang River Basin, Water Resources Department of
Gansu Province (http://www.gs.xinhuanet.com/shiyanghe/index.htm), the evaporation
loss of the entire Shiyang River Basin can be estimated.
$$f = \frac{\sum f_i \cdot V_i}{V} \tag{2}$$
where $f$ is total evaporation loss, $f_i$ is evaporation loss of each section, $V_i$ is the runoff
of each section (for a reservoir, it represents the storage capacity), and $V$ is the total
runoff.
**4. Results**
**4.1 Temporal variation of stable isotopes in surface water of**
**river-lake continuum**
The stable isotopic composition of surface water in the Shiyang River shows



differences in different sections (Fig. 2, Table 1). From mountains to the oasis to the
desert area, stable isotopic values show a trend of gradual enrichment. Especially in
desert areas, due to the extremely arid environment and scarce vegetation cover, a
large amount of surface water is susceptible to evaporation, and stable isotopic values
are therefore quite enriched. From the perspective of time, there is a slight variation in
stable isotopic values of surface water in mountainous and oasis areas during the
sampling period. However, stable isotopic values of surface water in desert areas
change a lot with time going by. It can be seen that stable isotopes were enriched in
summer and depleted in spring and autumn of 2017, while stable isotopes do not show
the same variation tendency despite the same great fluctuations of stable isotopic
value in 2018. This is because the stable isotopic values of the lake water in Qingtu
Lake are not only affected by evaporation, but also by human activities. When the
water transfer period comes, the water discharged from Hongyashan Reservoir with
lower stable isotopic value flows into Qingtu Lake, making the stable isotopic values
of lake water lower. During the non-transportation period, the lake water evaporates
continuously under hot and dry climate conditions, so the stable isotopes are gradually
enriched.




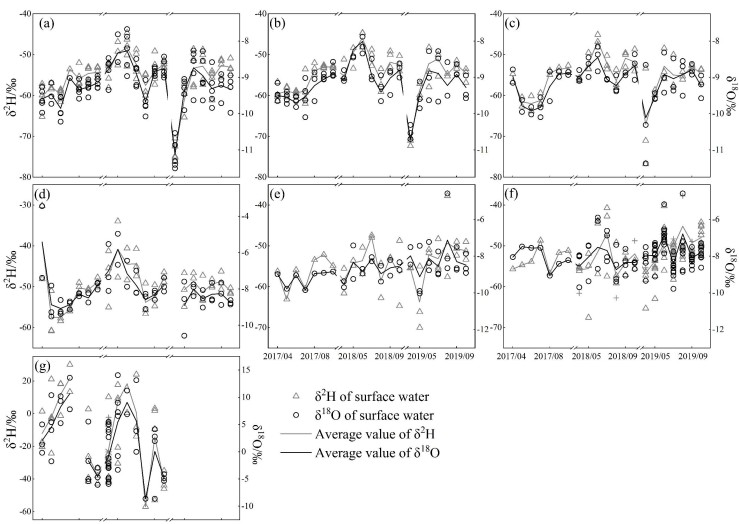


Fig. 2 Monthly variation of δ¹⁸O and δ²H in surface water of river-lake continuum during the
sampling period. (a): Mountain 1; (b): Mountain 2; (c): Xiying Reservoir; (d): Oasis 1; (e): Oasis 2;
(f): Hongyashan Reservoir; (g): Desert (Qingtu Lake).
Table 1 Characteristics of stable isotopic values of surface water at each sampling site during the
sampling period.

| Section | $\delta^{18}O$ (‰) | | | $\delta^2H$ (‰) | | |
|---|---|---|---|---|---|---|
| | Max. | Min. | Mean | Max. | Min. | Mean |
| Mountain 1 | -8.28 | -11.12 | -9.20 | -48.36 | -73.84 | -55.59 |
| Mountain 2 | -8.01 | -10.71 | -9.16 | -45.97 | -71.58 | -55.00 |
| Xiying Reservoir | -8.45 | -10.11 | -9.15 | -46.71 | -67.06 | -55.46 |
| Oasis 1 | -5.42 | -9.08 | -8.01 | -57.54 | -39.14 | -49.54 |
| Oasis 2 | -7.13 | -9.85 | -8.72 | -47.74 | -63.75 | -54.98 |
| Hongyashan Reservoir | -6.79 | -9.07 | -7.98 | -45.21 | -57.50 | -52.33 |
| Desert (Qingtu Lake) | 10.67 | -8.64 | 1.93 | 21.65 | -57.05 | -11.85 |

**4.2 Spatial variation of stable isotopes in surface water of river-lake**
**continuum**
Based on the dataset of stable isotopic composition analyzed from surface water,
we obtain average values of stable isotopes at each sampling point during the





sampling period. According to the distance between the sampling point and the river
source, a variation trend of stable isotopic values of surface water along the Shiyang
river is obtained, as seen in Fig. 3. Generally, with the flow of rivers, stable isotopes
show a trend of gradual enrichment, which is mainly attributed to evaporation. The
explanation may be rather the gradual increase of the degree of evaporation in the
river water along the river pathway. The "inflow water" for each section, is the isotope
composition of the preceding section, which becomes progressively enriched from
upstream to downstream.

In mountainous areas, altitude ranges from 2,000 to 5,000 m. In addition, due to

the climbing effect of air masses, there is more precipitation and higher vegetation
coverage in mountainous areas. Alpine shrubs dominate the vegetation coverage in the
upper reaches of the Shiyang River. Therefore, air humidity is higher due to the
transpiration of high-density plants and large precipitation. Both lower temperature
and higher humidity make the evaporation of surface water weaker in mountainous
areas. Thus, the enrichment of isotopes is not apparent in the mountainous areas, as
seen in Fig. 4. In oasis and desert areas, altitude ranges from 1,500 to 2,000m. The
natural vegetation coverage is low, mainly distributed on both sides of river banks and
roads. Especially at the end of the Shiyang River where the sandy soil has poor water
holding capacity and the climate is dry, there is very little vegetation. Relatively
higher temperatures and lower air humidity make evaporation severe (Sun et al.,
2021). As shown in Fig. 3, the stable isotopes of surface water are the most enriched
in the desert areas. However, there is a depletion between the sampling points O3 and
O4, which may be caused by water transfer across river basins. (Beginning in 2001,
Gansu Province implemented the Minqin Water Transfer Project to introduce 100
million m³ Yellow River water into the Shiyang River every year, and the water outlet
is located between the sampling points O3 and O4). The hydraulic connection
between Hongyashan Reservoir and Qingtu Lake is maintained by the water
conveyance channel, and this connection only exists during the water transfer period.
Therefore, the stable isotopes of surface water are quite enriched in Qingtu Lake than
Hongyashan Reservoir.

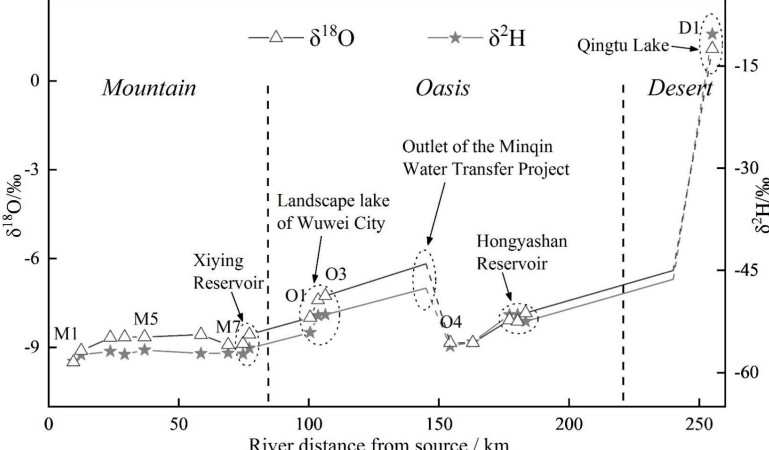


Fig. 3 Average values of stable isotopes in surface water collected at Shiyang river during the
sampling period, plotted according to distance from the source.
**4.3 Calculation of evaporation loss**
Evaporation loss of each section ranges from 0.85% to 39.88%, estimated by
$\delta^{18}O$, and from 1.61% to 42.86%, estimated by $\delta^2H$, respectively (Table 2). According

Hydrology and Earth System Sciences
Author(s) 2022



to the estimated evaporation loss, it can be found that whether it is estimated by $\delta^{18}O$
or $\delta^2H$, it gradually increases with the flow of the Shiyang River. There is only an
abnormal value in section oasis 2. Just as we found anomalies in isotope values
between sampling O3 and O4, this is mainly caused by water transfer across river
basins. Due to the impact of the Yellow River water transferred into the Shiyang River
Basin, the estimation results of the evaporation loss of this section may be biased. The
estimated evaporation loss of this section is lower than that of the adjacent upstream
section oasis 1. In addition,    the evaporation loss is significantly higher in large open
water areas such as Xiying Reservoir and Hongyashan Reservoir. In the upper and
middle reaches of the Shiyang River Basin, the impact of evaporation on water loss is
limited. However, in the Qingtu Lake of desert area, the water lost by evaporation
accounts for 41.37% of the water discharged from the Hongyashan Reservoir, and it
accounts for most of the evaporation loss in the Shiyang River Basin. Due to the hot
weather and extremely arid environment, the high evaporation loss is considered
reasonable in Qingtu Lake.

The annual average runoff of each section of Shiyang River is 337 million $m^3$ in

Mountain 1, 384 million $m^3$ in Mountain 2, 406 million $m^3$ in Xiying Reservoir, 130
million $m^3$ in Oasis 1, 299 million $m^3$ in Oasis 2, 393 million $m^3$ in Hongyashan
Reservoir, and 350 million $m^3$ in Desert (Qingtu Lake), respectively. Based on Eq. 2,
the evaporation loss of mountainous river (including the sections of Mountain 1 and
Mountain 2) is estimated to 1.30%, the evaporation loss of oasis river (including the



sections of Oasis 1 and Oasis 2) is estimated to be 2.87%, and the total evaporation
loss of river-lake continuum in Shiyang River Basin is estimated to 14.66%, which is
a major loss for the Shiyang River Basin where water resources are already scarce.
Table 2 Comparison of evaporation loss calculated from changes in stable isotopes flowing into
and out of each section in Shiyang River.

|  | $\delta^{18}$O-based (%) | $\delta^2$H-based (%) | Mean (%) |
|---|---|---|---|
| Mountain 1 | 0.85 | 1.63 | 1.24 |
| Mountain 2 | 1.11 | 1.61 | 1.36 |
| Xiying Reservoir | 2.01 | 2.55 | 2.28 |
| Oasis 1 | 2.27 | 3.86 | 3.06 |
| Oasis 2 | 2.02 | 3.54 | 2.78 |
| Hongyashan Reservoir | 7.02 | 8.93 | 7.97 |
| Desert (Qingtu Lake) | 39.88 | 42.86 | 41.37 |
| Shiyang River Basin | 13.55 | 15.75 | 14.66 |

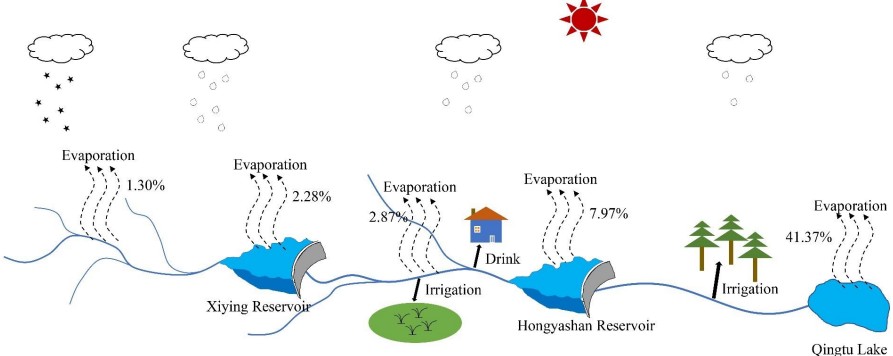


Fig. 4 A schematic of the Shiyang River, showing the source and consumption of river water and
the strength of evaporation on different sections.

## 5. Discussions

### 5.1 Difference in evaporation loss estimation based on $\delta^{18}$O and $\delta^2$H

Compared with the GMWL, the Local Meteoric Water Line (LMWL) is
constructed from the stable isotopes of hydrogen and oxygen in local precipitation,



which would better describe local meteorological conditions. Similarly, SWL is fitted
by $\delta^{18}O$ and $\delta^2H$ of surface water in Shiyang River (Fig.5). It plays an important role
in revealing the isotopic composition of local surface water and the strength of its
evaporation. Except for section mountain 1, the slopes of SWL in other sections are
all lower than that of GMWL and LMWL. Theoretically, the slope is lower than 8
because evaporation is driven by equilibrium and kinetic isotope effects (Craig, 1961;
Kattan, 2008). The lower the slope, the lower relative humidity, the higher the
evaporation rate. Whether surface water is transformed from groundwater or directly
recharged by precipitation, it will be affected by evaporation during the replenishment
process and the flow process (Cui et al., 2017). Therefore, the stable isotopes of the
river water have changed, and the heavy stable isotopes are gradually enriched in the
residual water with the increase of the downstream distance. As shown in Fig. 5, the
slope of the SWL of each section shows a gradually decreasing trend with the flow of
the Shiyang River. (Sun et al., 2021) have also found the rule of the increasingly
lower slope of SWL with the decrease of elevation under the control of temperature
and relatively humidity.

According to the estimation results in Table 2, it can be seen that the evaporation

loss estimated based on $\delta^2H$ is inconsistent with the evaporation loss estimated based
on $\delta^{18}O$, and the results based on $\delta^2H$ are significantly larger than that based on $\delta^{18}O$.
Most previous studies focus on evaporation loss of sizeable open water in a relatively
stable state, while this work presented here focuses on evaporation loss of rivers in a



flowing state, which may lead to a difference in evaporation loss estimated based on
$\delta^2H$ and $\delta^{18}O$. In addition, this difference may also be caused by the inconsistency
between the sampling schedule and the flow velocity of river water (Chen and Tian,
2021), resulting in the evaporation loss estimation affected by the possible temporal
change in source water isotopes. (Wu et al., 2017) point out that precipitation could
also cause this difference. If rainfall occurs during the sampling period, the rainwater
mixes into the river water, causing inconsistency in the samples collected upstream
and downstream. Therefore, sampling should be carried out during the raining-free
period to eliminate the impact of precipitation on the river water. This difference may
also be caused by ambient vapor isotopes (Mayr et al., 2007). In this work, ambient
vapor isotopes are derived from precipitation isotopes, which may differ from the
isotopes in the actual ambient vapor, reducing the representative of the ambient vapor
isotopes above the river. The reason for this difference may also come from the
influence of the averaging of meteorological parameters used in the *Hydrocalculator*.
Moreover, this difference seems to be common in evaporation loss estimation based
on $\delta^{18}O$ and $\delta^2H$ (Gibson et al., 1993; Haig et al., 2020).

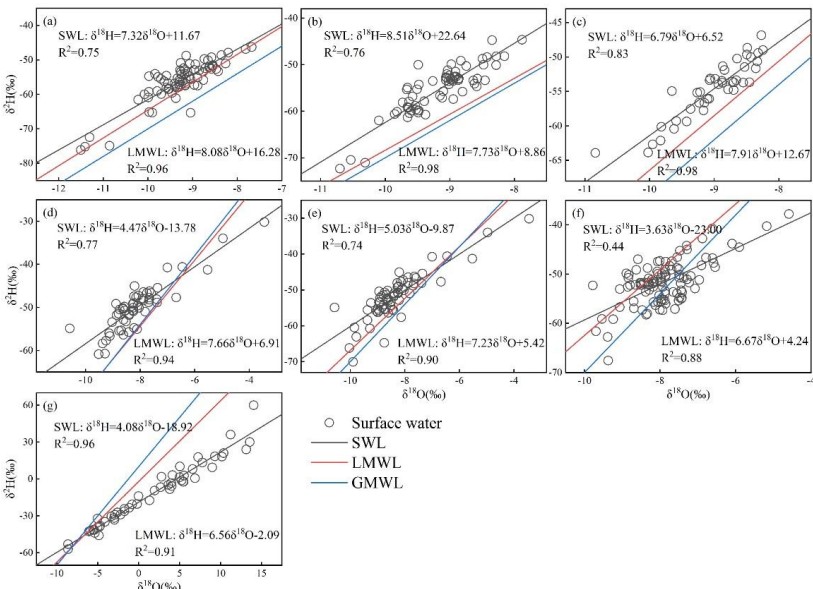


Fig. 5 $\delta^{18}O$ and $\delta^{2}H$ relationship for the surface water samples collected in the 7 sections and their

corresponding LMWL and SWL. (a): Mountain 1; (b): Mountain 2; (c): Xiying Reservoir; (d):

Oasis 1; (e): Oasis 2; (f): Hongyashan Reservoir; (g): Desert (Qingtu Lake).

## 5.2 Uncertainties

There may be several possible issues with the evaporation estimation in this

work. These issues mainly include sampling design, conceptual model for the flow of

water in the Shiyang River system, and calculation of related input parameters.

### 5.2.1 Water conservancy projects

Research has proved that water conservancy projects such as reservoirs play an

important role in shaping river water parameters (Grabowska, 2012). The construction

of various water conservancy projects on the Shiyang River and agricultural irrigation

will affect our estimation of the evaporation results. In the Shiyang River Basin, a

series of hydropower stations and reservoirs have been built for flood control and



irrigation, and Xiying Reservoir and Hongyashan Reservoir are the typical ones (Fig.
1). When the runoff is adjusted by water conservancy engineering facilities, it will
inevitably cause the flow velocity to be different from the natural state under the
conditions of manual intervention. For example, before the dry season, the discharge
of water will be reduced in response to a possible later drought, and the downstream
flow velocity will slow down due to the reduction of the discharge of water, and the
same is true before the rainy season. If our sampling schedule coincides with the
runoff adjustment of the water conservancy project, this will definitely affect the
evaporation estimation. Therefore, we will avoid the reservoir storage adjustment
period before making a sampling schedule.
Although there is manual intervention in the runoff, the input and output of water
are kept in a relatively stable state for a water conservancy project. However, water
conservancy projects would also cause water retention (Wang et al., 2019). After
flowing through the water conservancy projects, the outflowing water comes from the
deep layer of reservoirs, and the stable isotopes of the deep layer are more enriched
than those of the surface layer, which may affect the evaporation estimation to a
certain extent (Fig. 6). Therefore, to reduce the impact of water retention caused by
water conservancy projects on the evaporation estimation, they are considered a
separate section when estimating river evaporation. For example, we estimate the
evaporation loss of Xiying Reservoir and Hongyashan Reservoir in this work,
respectively.





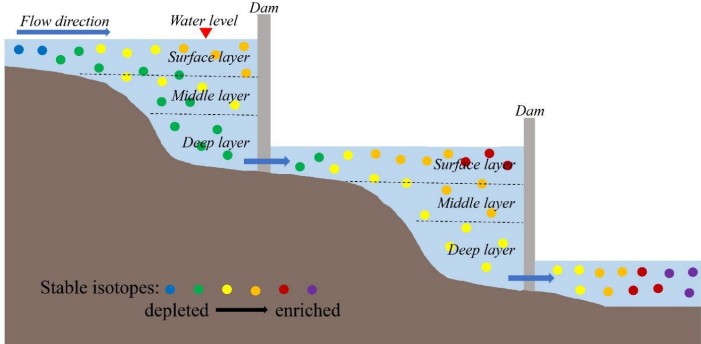


Fig. 6 A schematic describing the effect of dams on the distribution of stable isotopes in surface

water of river-lake continuum.

**5.2.2 Non-in-stream-evaporative causes of isotope changes**

*Hydrocalculator* is proposed based on the C-G model. An important premise of

the C-G model is that there is no input from other water bodies in the study area (i.e.,
no additional surface water, groundwater, or rainfall inflow during the study period)
(Skrzypek et al., 2015). However, this is an ideal state. In fact, groundwater and
surface water constantly recharge each other in the process of river flows, and this
process is difficult to quantify with current technology. The continuous mutual
transformation between groundwater and surface water causes constant changes in the
stable isotopic values of surface water. Ultimately, the water we collect downstream is
not only enriched by evaporation, but also enriched or depleted by the inflow of
groundwater. However, there are large uncertainties in quantifying the conversion
volume of surface water to groundwater, so it is difficult to calculate the extent of its
influence on the evaporation estimation of the Shiyang River.

Equally challenging to quantify are the irrigation return flows from the vigorous





agricultural activity in the middle and lower reaches of the Shiyang River. The Wuwei
and Minqin Oasis, located in the middle and lower reaches of the Shiyang River Basin,
are one of the most important grain bases in the Hexi Corridor. Agricultural activities
are vigorous in this area. However, the booming agrarian production inevitably
requires many water resources (Jia et al., 2020). Flood irrigation is the main irrigation
method in this area. This method not only causes the loss of a large number of water
resources due to evaporation, but also increases the return of irrigation water. This
infiltration may have a significant cumulative effect on river flow and stable isotopic
composition (Yoshida et al., 2016).
**5.2.3 Model sensitivity analysis**
The estimated evaporation loss exhibits an increasing trend with the increase of
temperature (T), relative humidity (h) and isotopic values of outflow water ($\delta_Q$), while
shows a decreasing trend with the increase of isotopic values of precipitation ($\delta_R$) (Fig.
8). In addition, the response of evaporation loss to different variables differed
substantially. Among those variables, relative humidity is the factor that has the
greatest impact on evaporation loss estimation, followed by stable isotopes of
precipitation and temperature (Fig. 7). The evaporation loss increased most steeply
with high relative humidity conditions (Fig. 7cd). The relative humidity significantly
influences evaporation flux over the surface water and determines the isotope kinetic
fractionation. As shown in Fig. 7cd, the estimated evaporation loss increases sharply
at a given isotopic value when the relative humidity changed by 10%, but increases



rapidly under conditions of high humidity (h = 70%). Isotopic composition of
atmospheric water vapor ($\delta_A$) is the direct controlling factor of evaporation loss, but in
this work, $\delta_A$ is calculated from isotopic composition of precipitation ($\delta_R$) because it is
difficult to measure it directly in remote regions (Gibson, 2002; Li et al., 2021; Wu et
al., 2017). The high variability in $\delta_R$ possibly increases the uncertainty of the
calculated evaporation loss values. As observed in Fig. 8ef, an increase in $\delta^{18}O_R$ by
3‰ or $\delta^2H_R$ by 10‰ will lead to a decrease in evaporation loss at any given $\delta_Q$.
evaporation loss increased more sharply with an increase in $\delta_Q$ under low $\delta_R$ values. In
addition, although evaporation loss ratios change slightly with the variation of
temperature (Fig. 7ab), the temperature is also an important factor influencing the
evaporation flux over surface water (Kumar and Nachiappan, 1999). It determines the
isotopic fractionation at the interface between the surface water and vapor (Horita et
al., 2008), and affects the estimation of evaporation loss. This analysis indicates that
the main sources of uncertainties of evaporation loss are mainly derived from the h,
followed by $\delta_R$ and T.



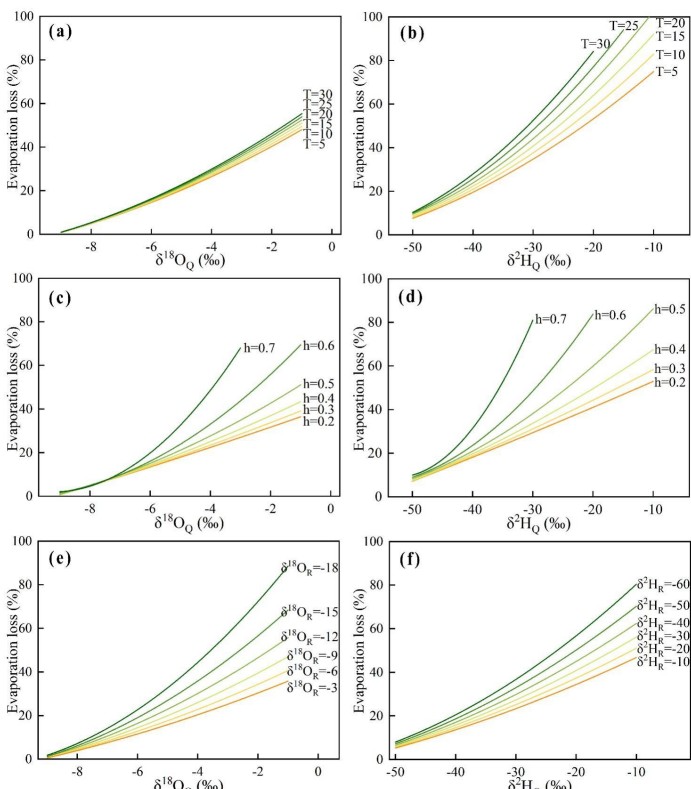

Fig. 7 The uncertainty assessment of E/I as the variations of input variables ($\delta^{18}O_Q$, $\delta^2H_Q$, $\delta^{18}O_R$,

$\delta^2H_R$, h, T).

## 5.3 Implications

The quantification of the water balance in the Shiyang River Basin provides
important insights into the hydrological processes in arid and semi-arid regions.
Although the annual precipitation is in short supply, agriculture is highly developed in
Shiyang River Basin, and agricultural water mainly comes from the extraction of
groundwater and the interception of river water. In order to intercept the incoming
water, a series of small and medium-sized reservoirs were built. There is no doubt that
the construction of the reservoirs has increased the water area and increased the water





loss due to evaporation in lower reaches where evaporation is initially strong, which is
consistent with our evaporation loss estimation. The evaporation loss of Xiying
Reservoir and Hongyashan Reservoir is relatively large, which are 2.28% and 7.97%,
respectively. For arid and semi-arid regions, how to reduce unnecessary waste of
water is an important part of the sustainable development of water resources. This
work could provide a management basis for water resources management departments.
In the next step, relevant research can be carried out to discuss how to reduce the large
amount of water evaporation caused by the construction of the reservoirs. For
example, transfer surface reservoirs to the ground (Ouerdachi et al., 2012), or cover
floating balls on the surface reservoirs (Rezazadeh et al., 2020).

Previous research on evaporation loss mainly focused on large open water bodies

such as lakes and reservoirs, while paying less attention to natural rivers. The
proposal of this work has potential application for the study of regional hydrological
cycles in other regions of the world. However, each river system has its own unique
features. The sampling design should conform to its characteristics and consider all
possible influence factors. For example, the impact of precipitation on river water is
greater in humid regions than that in arid regions. When estimating evaporation loss,
the impact of precipitation in the moist areas should be fully considered. In addition,
the sampling schedule should be as consistent as possible with the flow velocity. For
rivers with long distances, appropriate encryption sampling is required. If an
important tributary flows in, samples of the tributary should also be collected.





Moreover, long-term observations will make the results more reliable and convincing.

## 6. Conclusions

Research on the evaporation loss of the river-lake continuum in the Shiyang
River shows that stable isotopic technology can be used as a quantitative analysis tool
to reveal the water balance of a basin. We have conducted systematic observations of
stable isotopes of the river-lake continuum from the source to the end of the Shiyang
River for three years. The results show that stable isotopes in the river-lake continuum
gradually enriched from the source to the end of the Shiyang River. Besides, the stable
isotopes of river water show obvious seasonal variations, enriched in summer and
depleted in spring and autumn. Based on the *Hydrocalculator*, we estimate the
evaporation loss of each section of the Shiyang River. The evaporation loss of
Shiyang River is estimated to be 1.30% in the mountainous rivers, 2.28% in the
mountainous reservoir (Xiying Reservoir), 2.87% in the oasis rivers, 7.97% in oasis
reservoir (Hongyashan Reservoir), and 41.37% in the terminal lake (Qingtu Lake).
The estimation loss shows a trend of gradual increase from upstream to downstream.
No matter in mountain or oasis, the evaporation loss of reservoir is much higher than
that of the river. The largest evaporation loss appears in the section of Qingtu Lake at
the end of Shiyang River, with a value of 41.37%, which also explains the
disappearance of Shiyang River here. According to the water volume and evaporation
loss of each section, the evaporation loss of the entire river basin is calculated to be
about 14.66%.


Furthermore, we also observed the inconsistency of evaporation loss estimation
based on $\delta^{18}O$ and $\delta^{2}H$, which may be associated with the factors including
inconsistency of sampling schedule and flow velocity, rainfall during the sampling
period, representativeness of ambient vapor stable isotopes, and meteorological
parameters used in the model. Although there are some uncontrollable factors in
estimating natural river evaporation loss, this work provides a stable isotopic method
for assessing water volume variations in natural river basins.

## Acknowledgments:

This work was supported by the National Natural Science Foundation of China
[grant number: 41867030, 41971036]. The authors are very grateful to their
colleagues in the Northwest Normal University for their help in fieldwork, laboratory
analysis, data processing.

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
