# Peer review of "Correspondence Authors"

_Hydrology and Earth System Sciences, 2022_

## Referee Comment (RC1)

Dear authors of "Evaporation loss estimation of river-lake continuum of arid inland river: Evidence from stable isotopes", I had the pleasure to read your paper and in the following I provide some comments and suggestions aimed at improving your paper in a potential revision. I think that your paper has potential to advance the application and hydrological assessments using stable isotope tracing techniques to quantitatively estimate the evaporation loss from a flowing river system in the Shiyang River Basin, Northwest China. In this manuscript it is a key result that the evaporative loss of storage reservoirs (lakes, dams) and flowing river is calculated separately, which clarifies the priorities of future water management in arid and semi-arid regions and also would be a good addition to the field of water budgets in arid and semi-arid regions. Having said that, I have some suggestions you could consider incorporating into a revised paper:

Major comments:

1. Section 3.3: I would recommend adding more information on the calculations performed using the Hydrocalculator. I further encourage the authors to add a table showing the input parameters used for each section and explain in the main text how these values have been derived.

2. Section 5.1: The author should clarify the specific estimation error between $\delta D$ and $\delta^{18}O$. More evidence/calculations should be provided in terms of the errors.

3. Section 5.2.3: It's great to run sensitivity tests of the calculated evaporation loss to the model input variables. However, the authors only analyzed the sensitivity of temperature, humidity, outflow water and precipitation. Please add the sensitivity analysis of inflow water to the model.

4. The paper should also be thoroughly edited for language, as I detected many odd wordings and grammatical errors. Some sentences are not clear and sometimes convey incorrect and confusing messages for readers.

Minor comments:

1. Lines 127-132: The datasets or sources of temperature and humidity used in the main text should be clarified.

2. Line 146: the "b" of "basin" should be capitalized.

4. Line 188: When using the abbreviation (C-G), it should be marked at the first occurrence of its full name.

5. Line 262: "temperatures" should be "temperature".

6. Line 281, 286: the "o" of "oasis" should be capitalized.

7. Line 282: "sampling O3 and O4" should be " sampling points O3 and O4".

8. Lines 312: "SWL" refers to the surface water lines?

9. Line 315: the "m" of "mountain" should be capitalized.

10. Line 318: " the lower relative humidity " should be " the lower the relative humidity ".

11. Line 325, 337: Citation format is not corrected.

12. Line 337: "point" should be "pointed".

13. Line 377: "enriched" should be "depleted".

14. Line 395: It might be more appropriate to revise "the water we collect downstream" to "the water collected downstream".

15. Line 427: There is no Fig. 8 in the manuscript.

16. Line 429: the "e" of "evaporation" should be capitalized.

17. Line 437, 438: The abscissa title is inconsistent with the description of Fig. 7.

18. Line 450: "are" should be "is".

19. Line 483: "estimation" should be "evaporation".

---

## Author Comment (AC1)

**Title: Evaporation loss estimation of river-lake continuum of arid inland river: Evidence from stable isotopes**

**MS No.: hess-2022-75**

**MS type: Research article**

**Responses to the Reviewers:**

1. Section 3.3: I would recommend adding more information on the calculations performed using the Hydrocalculator. I further encourage the authors to add a table showing the input parameters used for each section and explain in the main text how these values have been derived.

**Reply:** Thank you for your valuable suggestion. We have added the details on parameters used in Hydrocalculator as follows:

Despite the complexity of the hydrological cycle in the Shiyang River Basin, the influence of other water (precipitation, groundwater and irrigation water) input is ignored in the calculation of evaporation loss for each section. Based on the stable isotope composition of inflow water and outflow water, evaporation loss of each section can be calculated:

$$\frac{E}{I} = \frac{\delta_L - \delta_P}{m(\delta^* - \delta_L)} \tag{2}$$

The evaporation loss of each section is calculated based on the *Hydrocalculator* software. Due to the constant inflow or outflow of river water in these sections, the steady-state model should be selected for the calculation of evaporation loss at each section. Therefore, we should select the value of columns  $EI_H$  and  $EI_O$  in the *output* file as the estimation results of the evaporation loss in these sections.

All parameters and their descriptions used in Hydrocalculator are listed in Table

| Parameters           | Parameter description                           |                           |
|----------------------|-------------------------------------------------|---------------------------|
| Т                    | Temperature (°C)                                |                           |
| h                    | Relative humidity (%)                           |                           |
| $\delta_{ m R}$      | Stable isotopes of precipitation (‰)            |                           |
| $\delta_{ m P}$      | Stable isotopes of inflow water (‰)             | Measured or assumed       |
| $\delta_{ m L}$      | Stable isotopes of outflow water (‰)            |                           |
| $\delta_{ m A}$      | Stable isotopes of air ambient moisture (‰)     |                           |
| Slope LEL | The slope of local evaporation line             |                           |
| $\mathcal{E}_K$      | Kinetic isotope fractionation factor (‰)        | Calculated from the model |
| $\mathcal{E}^+$      | Equilibrium isotope fractionation factor (‰)    |                           |
| З                    | Total isotope fractionation (‰)                 |                           |
| $C_k$                | The kinetic fractionation constant (‰)          |                           |
| $lpha^+$             | Equilibrium isotope fractionation factor (‰)    |                           |
| $\delta^{*}$         | Limiting isotopic composition (‰)               |                           |
| т                    | Isotope enrichment slope                        |                           |
| E/I_H                | Evaporation loss calculated based on hydrogen   |                           |
|                      | (%)                                             | Results                   |
| E/I_O                | Evaporation loss calculated based on oxygen (%) |                           |

Table 1 Parameters and their description involved in evaporation loss calculation.

2. Section 5.1: The author should clarify the specific estimation error between  $\delta D$  and  $\delta 18O$ . More evidence/calculations should be provided in terms of the errors.

**Reply:** Thank you for pointing this out. This error occurs in almost all similar studies, and the reasons for the error are multi-factors. At present, it is difficult to clearly analyze the specific influence of a single factor on the error of estimation results between  $\delta D$  and  $\delta^{18}O$ . Currently, we have added comparative studies in terms of sampling (adding the control group), experiment (adding the control group), and calculation (comparative verification of different calculation methods), hoping to minimize the uncertainty.

3. Section 5.2.3: It's great to run sensitivity tests of the calculated evaporation loss to the model input variables. However, the authors only analyzed the sensitivity of temperature, humidity, outflow water and precipitation. Please add the sensitivity analysis of inflow water to the model.

Reply: Thank you for your valuable suggestion. We have added the sensitivity

analysis of inflow water to the model and rewrite this section as follows:

The estimated evaporation loss (E/I) exhibits an increasing trend with the increase of temperature (T), relative humidity (h) and isotopic values of outflow water  $(\delta_L)$ , while shows a decreasing trend with the increase of isotopic values of precipitation ( $\delta_R$ ) and inflow water ( $\delta_P$ ) (Fig. 7). In addition, the response of evaporation loss to different variables differed substantially. Isotopic composition of atmospheric water vapor ( $\delta_A$ ) is the direct controlling factor of evaporation loss, but in this work,  $\delta_A$  is calculated from isotopic composition of precipitation ( $\delta_R$ ) because it is difficult to measure it directly in remote regions (Gibson, 2002; Li et al., 2021; Wu et al., 2017). The high variability in  $\delta_R$  possibly increases the uncertainty of the calculated evaporation loss values. As observed in Fig. 7ab, an increase in  $\delta^{18}O_R$  by 3‰ or  $\delta^2 H_R$  by 10‰ will lead to a decrease in evaporation loss at any given  $\delta_L$ . Evaporation loss increased more sharply with an increase in  $\delta_L$  under low  $\delta_R$  values. The deviation between  $\delta_P$  and  $\delta_L$  determines the value of E/I. As can be seen from Fig. 7cd, for a given  $\delta_P$ , the trends of E/I change parallel to each other as  $\delta_L$  changes, indicating that E/I shows a linear variation with both  $\delta_P$  and  $\delta_L$  for a given other parameters.

Temperature and relative humidity are the two most important meteorological factors controlling surface water evaporation. The relative humidity significantly influences evaporation flux over the surface water and determines the isotope kinetic fractionation. With other variables held constant, E/I increases continuously with increasing relative humidity (Fig. 7ef). Besides, evaporation loss increased most

steeply with high relative humidity conditions (h=70%) (Fig. 7ef). Although evaporation loss ratios change slightly with the variation of temperature (Fig. 7gh), the temperature is also an important factor influencing the evaporation flux over surface water (Kumar and Nachiappan, 1999). It determines the isotopic fractionation at the interface between the surface water and air (Horita et al., 2008), and affects the estimation of evaporation loss. In general, E/I is more sensitive to relative humidity variation than to temperature variation.

---

## Author Comment (AC2)

**Title: Evaporation loss estimation of river-lake continuum of arid inland river: Evidence from stable isotopes**

**MS No.: hess-2022-75**

**MS type: Research article**

**Responses to the Reviewers:**

**A simple application of the Hydrocalculator without details on parameter setups and assumptions on hydrology is not acceptable here. Hydrocalculator was developed for the purpose of easing isotopic mass balance calculation, but it still requires prior knowledge of hydroclimatology and regional isotopic signatures in the study site to be able to choose isotopic inputs with reasonable assumptions. To help readers understand and evaluate the results, details on parameters, (especially dI and dA) must be provided.**

**Reply:** Thank you for your valuable suggestion. We have added regional hydroclimatology in the section of 3.1 Sampling design. The content is as follows:

The middle and upper reaches of the Shiyang River are mainly natural rivers, while the downstream section from Hongyashan Reservoir to Qingtu Lake is mainly artificial water channels. Influenced by the confluence of tributary water system, groundwater surface water exchange and irrigation return flow, there are many uncontrollable factors in evaporation loss estimation from natural river channels. Therefore, the estimation of evaporation loss from this section was divided into several small sections. Since there are special hydrological processes in reservoirs, lakes and other water systems, evaporation loss from these water systems are analyzed separately. Based on the setting of our sampling sites in the Shiyang River basin and the water circulation of different sections, the Shiyang River was divided

into seven sections for evaporation loss calculation, including four runoff sections (Mountain 1, Mountain 1, Oasis 1, Oasis 2), two reservoir sections (Xiying Reservoir, Hongyashan Reservoir) and one lake section (Desrt (Qingtu lake)).

Besides, we have added the details on parameters used in Hydrocalculator as follows:

Despite the complexity of the hydrological cycle in the Shiyang River Basin, the influence of other water (precipitation, groundwater and irrigation water) input is ignored in the calculation of evaporation loss for each section. Based on the stable isotope composition of inflow water and outflow water, evaporation loss of each section can be calculated:

$$\frac{E}{I} = \frac{\delta_L - \delta_P}{m(\delta^* - \delta_L)} \tag{2}$$

The evaporation loss of each section is calculated based on the *Hydrocalculator* software. Due to the constant inflow or outflow of river water in these sections, the steady-state model should be selected for the calculation of evaporation loss at each section. Therefore, we should select the value of columns *EI_H* and *EI_O* in the *output* file as the estimation results of the evaporation loss in these sections.

All parameters and their descriptions used in this work are listed in Table 1.

Table 1 Parameters and their description involved in evaporation loss calculation.

| Parameters | Parameter description | |
| --- | --- | --- |
| $T$ | Temperature (℃) | |
| $h$ | Relative humidity (%) | |
| $\delta_R$ | Stable isotopes of precipitation (‰) | |
| $\delta_P$ | Stable isotopes of inflow water (‰) | Measured or assumed |
| $\delta_L$ | Stable isotopes of outflow water (‰) | |
| $\delta_A$ | Stable isotopes of air ambient moisture (‰) | |
| *Slope$_{LEL}$* | The slope of local evaporation line | |
| $\varepsilon_K$ | Kinetic isotope fractionation factor (‰) | Calculated from the |

| | | |
|---|---|---|
| $\varepsilon^+$ | Equilibrium isotope fractionation factor (‰) | model |
| $\varepsilon$ | Total isotope fractionation (‰) | |
| $C_k$ | The kinetic fractionation constant (‰) | |
| $\alpha^+$ | Equilibrium isotope fractionation factor (‰) | |
| $\delta^*$ | Limiting isotopic composition (‰) | |
| $m$ | Isotope enrichment slope | |
| E/I_H | Evaporation loss calculated based on hydrogen (%) | Results |
| E/I_O | Evaporation loss calculated based on oxygen (%) | |

**The manuscript uses equation 2 to scale-up station-based estimation of f to catchment-scale f. However, their weighting approach is not justified. First of all, it is not clear what is the unit of V used here (mm or m3)? I do not know if the contribution area plays a role. Secondly, the manuscript does not provide any justification for using V as weighing metrics? Why not use potential evaporation or precipitation as weighting metrics?**

**Reply:** Thank you for your valuable suggestion. We have added the unit (m³/a) of V and f. Potential evaporation or precipitation cannot be used as weighing metrics here, because the I of E/I means inflow water, the evaporation amount of each section could be obtained only knows the inflow amount of each section.

**The sensitivity/uncertainty analysis (section 5.2.3) could be significantly improved. First of all, the author missed dA in the sensitivity analysis, which is an important source of uncertainty as I aforementioned. Secondly, uncertainties are not only sensitivities as shown in Figure 7, it is the product of sensitivity and variability of input parameter. I agree with the author that E/I is sensitive to h, however, if the overall variability of individual parameters is considered, the impact on uncertainties may be different for input parameters. Providing systematic sampling on dP, and numerous weather stations, the combination of sensitivities and parameter variabilities shall be evaluated together to assess uncertainties.**

**Reply:** Thank you for your valuable counsel. In this work, δA (Air ambient moisture) is not obtained by direct measurement, while it is estimated based on δR (Precipitation). Nevertheless, the estimation of δA is also corrected by LEL, which

has been widely used in those remote area with difficulties in directly measuring $\delta A$ and has been considered reasonable by numerous researches. Sensitivity analysis for $\delta A$ was not performed, but we did sensitivity analysis for $\delta R$.

**Line specific comments:**

**1. P4, lines 71-73 what is "constant climate change"?**

**Reply:** In order to correctly express the message, we have deleted the word "constant" and revised the sentence to "Therefore, as the main water resources in arid and semi-arid regions, it is necessary to systematically assess the evaporation loss of river systems in the context of climate change."

**2. P5, line 79, what is "mobile water system"?**

**Reply:** We are sorry for conveying the confused information. We have changed "mobile water system" to "the flowing water system" and revised the sentence to "Numerous studies have successfully estimated the evaporation loss of the large open waters using stable isotopes (Cui et al., 2017; Hernández-Pérez et al., 2020; Yapiyev et al., 2020), while few studies have focused on the flowing water system, such as natural rivers (Diamond and Jack, 2018) and artificial waterways (Chen and Tian, 2021)."

**3. P5, lines 80-87 I disagree with this statement! What do the authors mean by "instability factors"? These listed factors also impact isotopic application in lakes, wetlands, soils and reservoirs….**

**Reply:** This suggestion is appreciated. The original sentence only expresses in general terms all the possible uncertainties in evaporation loss estimation and does not highlight the differences in the flowing water systems. Therefore, we have revised the original sentence to "It may be that the flowing water system brings the biggest uncertainty: the possible inconsistency between sampling schedule and flow velocity (Chen and Tian, 2021), especially the water discharge and store of reservoirs in upstream will lead to changes in the flow velocity of downstream river (Luc and Bernhard, 2007; Aravena and Suzuki 1990)."

**4. P5 lines 89-90 What do the authors mean by "surface river system is the**

**main contributor to the surface evaporation"?**

**Reply:** What we want to express is that the river-lake continuum is the most important water resource on the surface, and its evaporation contributes a significant amount of evaporative flux from the land surface. To more accurately convey what we want to express, we have revised the original sentence to "Although there are some uncontrollable factors in the estimation of evaporation loss of the flowing water system, it is necessary to carry out relevant research because evaporation of the river-lake continuum is the main contributor to the surface evaporative flux (Chen and Tian, 2021)."

**5. P5 line 92, "production and life of local residents" is not the right expression**

**Reply:** We have changed the word "life" to "living" and revised the original sentence to "As one of the most important inland rivers in Northwest China, the Shiyang River plays an important role in the production and living of local residents and social development."

**6. P6 line 113, "a peak of 1300-2000 m" is not the right expression. May be "with the elevation in the range of 1300-2000 m"?**

**Reply:** Thank you for your significant reminding. According to your suggestion, we have revised the sentence "a peak of 1300-2000 m" to "with the elevation in the range of 1300-2000 m".

**7. P6 lines 118-119, Change "Precipitation" to "Precipitation amount"; "greater" to "great"**

**Reply:** Thank you for pointing this out. We have changed "Precipitation" to "Precipitation amount" and "greater" to "great", respectively.

**8. P6, lines 121-125 running sentence, please rephrase**

**Reply:** Thank you for your suggestion. We have rephrased this sentence to "Besides, the seasonality of precipitation in this area has a high degree of spatial heterogeneity, generally decreasing along the Shiyang River pathway with the mean annual precipitation amount reaches 700 mm in the southern mountainous region and less than 100 mm in the northern desert region (Sun et al., 2021)."

**9. P8, line 146, correct "Shyiang" to "Shiyang"**

**Reply:** Thank you for pointing this out. We have changed the word "Shyiang" to "Shiyang".

**10. P8 line 150 remove "more"**

**Reply:** We have deleted the word "more".

**11. P8 line 152, wonder why "natural river" is emphasized here?**

**Reply:** Thank you for pointing this out. We have revised the sentence to "5 surface water observation stations were also set up in the oasis section of Shiyang River".

**12. P9, line 168, no precipitation amount measured at weather stations?**

**Reply:** Thank you for pointing this out. The description of precipitation amount monitoring was forgotten in the initial writing. We have added it to the description of weather station parameter measurements and revised the sentence to "These stations record meteorological data every 30 minutes, including temperature, relativity humidity, dew point temperature and precipitation amount".

**13. P11 line 209, unit for f and V, please**

**Reply:** Thank you for your significant reminding. We have united the f and V and revised the sentence to "where $f$ is total evaporation loss of Shiyang River Basin (%), $f_i$ is evaporation loss (E/I) of each section (%), $V_i$ is the annual runoff of each section ($m^3/a$), and $V$ is the total annual runoff of Shiyang River Basin ($m^3/a$)".

**14. P14, line 246, "a trend of gradual enrichment" is not an accurate way to descript the observed pattern in Figure 3**

**Reply:** Thank you for your valuable counsel. Indeed, stable isotopes of river water tend to fluctuate greatly due to anthropogenic influences. Therefore, we have revised the sentence to "In general, stable isotopes show a trend of gradual enrichment as the river flows in the absence of anthropogenic interference, which is mainly attributed to evaporation".

**15. Figure 3, I would suggest adding standard deviation bars in the Figure.**

**Reply:** Thank you for your suggestion. We have added standard deviation bars in the Fig. 3.

[Figure]

Fig. 3 Average values of stable isotopes in surface water collected at Shiyang river during the

sampling period, plotted according to distance from the source.

**16. P15, lines 265-268 was the water transfer conducted seasonally or year-round?**

**Reply:** The water transfer conducted almost year-round. The amount of water transferred may vary seasonally, and the management department will adjust the amount according to the changes in water demand. For example, during periods of high irrigation activity, increasing water demand leads to an increase in water transfer.

**17. P16 lines 294-297 how does the annual average runoff (in Mm3) come?**

**Reply:** The annual average runoff is obtained in the Water Resources Utilization Center of Shiyang River Basin, Water Resources Department of Gansu Province (http://www.gs.xinhuanet.com/shiyanghe/index.htm), which has been explained in the section of 3.3 Craig-Gordon model.

**18. P19 Lines 331-334, I do not agree with the statement**

**Reply:** This is only hypothetical and no definite conclusions can be drawn due to the fact that there are very limited researches on evaporation losses from flowing water bodies and literature comparisons cannot be carried out. However, there is no doubt that the uncertainty in the estimation of evaporation losses from flowing water

bodies is large, and it cannot exclude the influence of flowing water bodies on the evaporation loss differences based on hydrogen and oxygen.

**19. P20, line 359-361 Water conservation projects deserve a section in the study area**

**Reply:** Thank you for your suggestion. We have added an overview of the water conservancy facilities in the study area. The content is as follows: "Since the 1950s, the increase in arable land downstream led to a sharp rise in irrigation water demand. In order to intercept river water for irrigation, 13 small and medium-sized reservoirs with a total capacity of 90 million cubic meters were built in the upstream of Shiyang River, while the Asia's largest desert reservoir, Hongyashan Reservoir, with a capacity of 127 million cubic meters, was built in the downstream area (Ma et al., 2010). A series of hydraulic engineering facilities built in the Shiyang River basin have changed the original hydraulic and flow parameters of the river."

**20. P21 line 371-372, I do not exactly follow the argument! The statement is different from the illustration (Figure 6)**

**Reply:** Figure 6 is mainly an illustration as a description of the second paragraph of this section, which illustrates the effect of reservoirs on the stable isotopes of river water. The first paragraph of this section describes the impact of reservoirs on flow rate of downstream river during the periods of water discharge and water store. We do communicate with the management department in advance to ensure that our sampling schedule avoids the reservoir capacity adjustment period.

**21. Line 413, figure 8?**

**Reply:** Thank you for pointing this out. It should be Fig. 7 and we have changed the "Fig. 8" here to "Fig. 7".

**22. Lines 420-422 I do not follow the statement. Stable state vs flowing state? Any reference for it?**

**Reply:** Lines 420-422 describe the fluence of relative humidity on evaporation loss and the results obtained based on our analysis, and it does not need a reference.

**23. Line 425, shall "Li et al., 2021" be "Li et al., 2016"?**

**Reply:** Thank you for pointing this out. After we double-checked the references

in the full text, it was found that the reference cited here was missed. Missing reference has been added to the manuscript as follows: "Li C L, Shi K B, Yan X J, Jiang C L. Experimental Analysis of Water Evaporation Inhibition of Plain Reservoirs in Inland Arid Area with Light Floating Balls and Floating Plates in Xinjiang, China[J]. Journal of Hydrological Engineering, 2021, 26(2): 04020060."

**24. Line 433, "the interface between the surface water and vapor"? Would be "between surface water and air"?**

**Reply:** Thank you for your valuable counsel. We have revised the original sentence to "It determines the isotopic fractionation at the interface between the surface water and air".

**25. Lines 434-436, I do not agree with this statement. The sensitivity exercise (Figure 7) only demonstrates that E/I results are more sensitive to h, than T, but it does not support the statement that the main source of uncertainties is h, because we did not know the overall variability of h, T, dA, while numerous weather stations are available to the study.**

**Reply:** Thank you for pointing this out. We have reorganized the logic of this section and rewritten the content. The content is as follows:

The estimated evaporation loss (E/I) exhibits an increasing trend with the increase of temperature (T), relative humidity (h) and isotopic values of outflow water ($\delta_L$), while shows a decreasing trend with the increase of isotopic values of precipitation ($\delta_R$) and inflow water ($\delta_P$) (Fig. 7). In addition, the response of evaporation loss to different variables differed substantially. Isotopic composition of atmospheric water vapor ($\delta_A$) is the direct controlling factor of evaporation loss, but in this work, $\delta_A$ is calculated from isotopic composition of precipitation ($\delta_R$) because it is difficult to measure it directly in remote regions (Gibson, 2002; Li et al., 2021; Wu et al., 2017). The high variability in $\delta_R$ possibly increases the uncertainty of the

calculated evaporation loss values. As observed in Fig. 7ab, an increase in $\delta^{18}O_R$ by 3‰ or $\delta^2H_R$ by 10‰ will lead to a decrease in evaporation loss at any given $\delta_L$. Evaporation loss increased more sharply with an increase in $\delta_L$ under low $\delta_R$ values. The deviation between $\delta_P$ and $\delta_L$ determines the value of E/I. As can be seen from Fig. 7cd, for a given $\delta_P$, the trends of E/I change parallel to each other as $\delta_L$ changes, indicating that E/I shows a linear variation with both $\delta_P$ and $\delta_L$ for a given other parameters.

Temperature and relative humidity are the two most important meteorological factors controlling surface water evaporation. The relative humidity significantly influences evaporation flux over the surface water and determines the isotope kinetic fractionation. With other variables held constant, E/I increases continuously with increasing relative humidity (Fig. 7ef). Besides, evaporation loss increased most steeply with high relative humidity conditions (h=70%) (Fig. 7ef). Although evaporation loss ratios change slightly with the variation of temperature (Fig. 7gh), the temperature is also an important factor influencing the evaporation flux over surface water (Kumar and Nachiappan, 1999). It determines the isotopic fractionation at the interface between the surface water and air (Horita et al., 2008), and affects the estimation of evaporation loss. In general, E/I is more sensitive to relative humidity variation than to temperature variation.

**26. Line 448 – "initially strong"?**

**Reply:** Thank you for pointing this out. We have changed the word "initially" to "originally" and revised the sentence to "There is no doubt that the construction of the reservoirs has increased the water area and increased the water loss due to evaporation

in lower reaches where evaporation is originally strong, which is consistent with our evaporation loss estimation".

**27. Line 467, Encryption sampling? what is it?**

**Reply:** Thank you for your significant reminding. It should be "intensive sampling", and the original sentence has been revised to "For rivers with long distances, appropriate intensive sampling is required".